# Wearable Sensors as a Preoperative Assessment Tool: A Review

**DOI:** 10.3390/s24020482

**Published:** 2024-01-12

**Authors:** Aron Syversen, Alexios Dosis, David Jayne, Zhiqiang Zhang

**Affiliations:** 1School of Computing, University of Leeds, Leeds LS2 9JT, UK; 2School of Medicine, University of Leeds, Leeds LS2 9JT, UK; a.dosis@leeds.ac.uk (A.D.); d.g.jayne@leeds.ac.uk (D.J.); 3School of Electrical Engineering, University of Leeds, Leeds LS2 9JT, UK; z.zhang3@leeds.ac.uk

**Keywords:** wearable sensors, exercise testing, preoperative assessment, perioperative pathway

## Abstract

Surgery is a common first-line treatment for many types of disease, including cancer. Mortality rates after general elective surgery have seen significant decreases whilst postoperative complications remain a frequent occurrence. Preoperative assessment tools are used to support patient risk stratification but do not always provide a precise and accessible assessment. Wearable sensors (WS) provide an accessible alternative that offers continuous monitoring in a non-clinical setting. They have shown consistent uptake across the perioperative period but there has been no review of WS as a preoperative assessment tool. This paper reviews the developments in WS research that have application to the preoperative period. Accelerometers were consistently employed as sensors in research and were frequently combined with photoplethysmography or electrocardiography sensors. Pre-processing methods were discussed and missing data was a common theme; this was dealt with in several ways, commonly by employing an extraction threshold or using imputation techniques. Research rarely processed raw data; commercial devices that employ internal proprietary algorithms with pre-calculated heart rate and step count were most commonly employed limiting further feature extraction. A range of machine learning models were used to predict outcomes including support vector machines, random forests and regression models. No individual model clearly outperformed others. Deep learning proved successful for predicting exercise testing outcomes but only within large sample-size studies. This review outlines the challenges of WS and provides recommendations for future research to develop WS as a viable preoperative assessment tool.

## 1. Introduction

Demand for general surgery is expected to increase in line with population ageing [1]. For various cancer types, abdominal surgery is considered first-line treatment [2,3]; for advanced-stage cancers, surgical treatment when combined with neoadjuvant treatment may be the only possible cure [2,3,4]. A common example of this is bowel/colorectal cancer, globally the third most common cancer [5]. Survival from bowel cancer has seen significant improvements in recent decades, with developments in surgical and perioperative care being suggested as reasons [6,7]. Further, mortality in the postoperative period has seen international decline and is frequently reported at below 5% [8,9,10]. However, postoperative complication rates in these populations have not seen the same decline.

Postoperative complications following abdominal surgery are frequently reported; up to a third of patients report some form of complication [11,12]. Common complications include surgical site infection, cardio-respiratory complications, gastrointestinal (GI) motility problems and anastomotic leak [13]. Postoperative complications have a profound impact not only on patients’ quality of life but also on the hospital as a service provider [13]. They are strongly associated with prolonged length of hospital stay, readmission to ICU and hospital re-admissions [13]. One large cohort review concluded that 23.3% of patients undergoing colorectal surgery were readmitted to hospital [14]. This results in a huge economic burden and has been evidenced widely across many studies [15]. With the population ageing, these costs are expected to increase up to 10% each year [16]. The identification of high-risk patients poses a crucial challenge for healthcare providers to optimise their allocation of resources and for the perioperative management of the patient. Preoperative assessment tools are used to support risk stratification of patients; however, none currently provide a precise assessment that is accessible to all patients.

### 1.1. Preoperative Assessment

Preoperative assessment occurs during the first stage of the perioperative period, as seen in Figure 1. Here, preoperative measurements are recorded from patients and used to stratify them into risk groups. The major goal of this is to identify the patients at the highest risk of perioperative morbidity and mortality [17]. As explained previously, it can also support the healthcare provider in resource allocation by estimating the support a patient may require across the perioperative pathway. However, as outlined by NICE guidelines, excessive preoperative testing is related to patient anxiety and significant delays to treatment [18]. Therefore, the benefits of testing should be carefully considered before implementation. The most common preoperative assessment tools are outlined here (see Figure 2).

Physical examinations build on an assessment of the patient’s medical history. These pre-anaesthesia examinations include a physical assessment of the lungs, heart function and possible evaluation of the main vital signs using a variety of tests [17,18]. NICE guidelines provide a breakdown of recommendations for testing that vary depending on the severity of the surgical treatment and health status of the patient [18]. For example, an ECG is a common tool that has been shown to optimise risk stratification of cardiovascular complications for non-cardiac surgery [21]. However, for minor/intermediate treatments in young or patients considered healthy, a resting ECG is not recommended as part of routine preoperative assessment [18]. These physical assessments are routinely performed at a preoperative clinic appointment in a resting state.

Multiple preoperative assessment tools exist that calculate patients’ risk of adverse outcomes from routinely collected data. These tools are widely recommended by medical societies to be employed as a preoperative assessment tool [22]. The ASA system (American Society of Anaethesiology), the APACHE II (Acute Physiology and Chronic Health Evaluation) and the POSSUM (Physiological and Operative Severity Score forenUmeration of Mortality and morbidity) scores have been shown to have good predictive value [23,24,25]. A review of common preoperative tools has shown that they have comparable predictive performance to machine learning (ML) techniques [26]. However, these tools are not consistently employed in practice. Lack of time and trust in the accuracy of measurements has frequently been reported by clinicians as a barrier to the implementation of risk calculators [27]. They present issues in that they can be open to subjectivity and sometimes require the input of variables that are not available in the preoperative period [25,28]. Further, the majority were originally developed with evidence that predates the last three decades of research [23,24,25].

Functional capacity assessment is a measure that aims to quantify the ability of a patient to undertake activities from a free-living environment that need ’sustained aerobic metabolism’ [29]. Much research has identified the association between a higher functional capacity and a reduction in postoperative complications [30,31,32]. The 6-min walk test is a common exercise tolerance test but lacks accuracy [33]. In comparison, cardiopulmonary exercise testing (CPET) is considered the current gold standard for preoperative assessment. CPET is a non-invasive clinical tool that evaluates cardio-respiratory function to measure exercise capacity [34]. In the clinic, patients undergo an exercise test on a cycle-ergometer or a treadmill whilst ventilation and respiratory gas parameters are measured [35]. Multiple studies have been shown to support its use as a tool to identify patients at increased risk of developing postoperative complications following general surgery [36,37]. Although CPET is a proven tool for risk stratification and is routinely implemented, there are several barriers to CPET being accessible and precise. CPET requires trained specialists to complete testing with ready access to dedicated facilities; in 2018, only 53% of Trusts in the UK offered the service [38]. Further, CPET is an expensive test with costs estimated at £289 per unit of testing in 2018/2019 (NHS Improvement, 2019). Although the test measures direct oxygen consumption, there can be considerable subjectivity with one study reporting possible miss-classification of outcomes in up to 60% of tests completed [39]. Finally, CPET might be contraindicated meaning that patients who are at high risk of complications are not always able to achieve a representative score, or even complete the test [40]. Wearable technology has been proposed as a tool that can overcome some of the barriers that are common in these preoperative assessment tools.

### 1.2. Wearable Sensors

The development of technologies in wearable sensors (WS) in the last decade has led to a significant increase in consumer uptake [41]. As a result, there is a vast quantity of data relating to individuals’ health whilst in free-living environments. There are also many examples of WS being implemented in a clinical setting as a cost-effective tool to measure physiological signals [42]. There is evidence to suggest that these devices could hold a vast volume of data that can give clinicians a quantitative representation of patients’ health in their day-to-day environment [43].

WS have multiple attributes that make them a suitable tool for preoperative assessment. Physiological signals have inherent biological variation and therefore, recording these signals over a longer time period may allow detection of abnormalities that present at irregular time periods [44,45]. A further advantage of collecting data over a longer time period is that the data may be more representative of normal routines. When collecting physical activity data, an increase in the number of days recorded is associated with a more reliable weekly estimate [46,47]. WS are usually autonomous devices that can record signals away from the clinic. This can provide a simpler alternative for clinicians who, under time constraints, cannot always complete preoperative physical assessments [27,48]. In some cases, measurements recorded away from the clinical environment may be more accurate; the ’White-coat’ effect describes the increases in physiological measurements that are only seen when taking measurements in the clinical environment [49]. These attributes have led to multiple successful implementations of WS for diagnostics.

WS have high efficacy for continuous monitoring of numerous physiological variables [50]. Subsequently, this has been shown to be applicable to support the diagnosis of several diseases including Parkinson’s, kidney failure and viral infections [51,52,53]. Particularly in the case of cardiovascular disease monitoring, WS can provide live monitoring capabilities of patients’ medical status that can be used to alert clinicians [54,55]. In the postoperative period, wearable technology has shown consistent uptake and to be a particularly useful tool for monitoring recovery [56]. WS have been used to identify post-surgical cancer patients who are recovering slower than their predicted profile, this facilitates the determination of appropriate discharge dates and preventing re-admissions [57]. A wide range of wearable sensing technology has been shown to be useful for clinicians in the postoperative setting including chest patches and wrist-based fitness sensors [58,59,60,61]. In the preoperative period, initial research has reported similar successful applications of WS.

Some research reports utilising WS as a method to measure adherence to prehabilitation programmes rather than as a preoperative risk assessment tool [62,63]. In other cases, WS have been utilised specifically for preoperative assessment with an exclusive focus on accelerometer data [64,65]. More recently, there have been instances where research has combined Heart Rate (HR) data with accelerometer data to approximate outcomes of cardiovascular fitness testing [40,66]. This research suggests the potential for advanced computing methods to analyse a combination of HR and movement data. These papers highlight the variation in sensor modalities and analysis methods that exist across the field. This suggests that the field would benefit from a review of these factors.

### 1.3. Aims of Review

Wearable sensors are a widely researched tool across medicine and exercise science. They have been shown to provide accurate and representative measurements that can give an objective insight into an individual’s health. In the postoperative period, multiple reviews have been conducted to provide an overview of WS both in the hospital and in the outpatient setting [56,67]. However, there is no review of research utilising WS in the preoperative period. This paper aims to investigate how WS have been used in free-living environments in research to predict either preoperative measurements or postoperative outcomes.

Research investigating emergency surgery will not be included as preoperative evaluation for emergency surgery does not allow analysis of free-living data. Additionally, the review will focus on major abdominal surgery; research completed in a cardiac or orthopedic surgical setting will be excluded as these procedures are associated with unique complications that are not applicable to a wider range of medical contexts [68]. Given that preoperative risk assessments are a vital component in the perioperative pathway, this paper will include research that has attempted to predict clinical variables that are routinely collected at this stage. Although this is not a systematic review, search terms were employed to identify research from selected databases; a narrative synthesis was used to summarise findings. The findings of this review are presented across four sections: Sensor Modalities, Pre-processing, Feature Extraction and Predictive Models.

### 1.4. Literature Search

To identify key literature and ensure valuable research was not missed, search terms were identified and used to search major databases. MEDLINE (Ovid) and Web of Science were searched as well as the first 200 returns in Google Scholar [69]. Relevant papers from ARXiv were also included. Search terms can be seen in Table 1. Further articles were identified through backward chaining.

Searches brought back a broad range of papers from which the most relevant were selected. Several inclusion criteria were outlined for the search. Research should analyse free-living data collected from WS in research. Data should be used to investigate the association of these signals with outcomes related to either clinical variables routinely collected preoperatively or postoperative outcomes. Clinical variables collected preoperatively varied from CPET outcomes to cardiovascular responses. A subsection of these papers was selected for in-depth analysis whilst multiple papers outside of this subset are referenced throughout this review. A compilation of the key features extracted from these papers selected for in-depth analysis can be found in Appendix A. A summary of the sample sizes and participant demographics can be seen in Appendix A. Sample sizes varied greatly across studies and had a range of 16 to 80,137. For research that had a sample size of under 1000, the mean sample size was 48.9 indicating that the majority of research in this field has a relatively small number of participants. This is likely due to having to provide hardware to each participant included in the research study rather than having access to pre-compiled data sets. There was also a broad range in the sex split across research; there was a slightly higher prevalence of male participants with an average of 57% males but this was not significantly imbalanced. Ethnicity was rarely reported with under 30% reporting the ethnic breakdown of participants. When ethnicity was reported there were significant imbalances with largely white participants.

For research collecting data from patients waiting to undergo surgery, participants were commonly approached immediately after being enlisted for surgery or at the preoperative evaluation clinic [70,71,72,73,74,75]. It was also common for patients to wear the device right up until the date of operation [70,71,76,77,78,79]. For all research apart from two papers, the data collection from the WS device took place within 33 days prior to surgical treatment. In two cases, it was unclear how far in advance of treatment patients wore their WS [64,80].

## 2. Hardware/Sensing Technologies

This review identified a range of different devices that have been utilised in the research in the preoperative setting. This paper does not give an analysis of each commercial device but presents an overview of the different sensor modalities that they employ to collect data. The accelerometer, ECG and PPG sensors were widely implemented; Figure 3a shows the breakdown of these sensor modalities across research. A full breakdown of the WS devices and models along with their sensor modalities can be found in Appendix A. The following sections outline the functionality of these sensors and their outputs.

### 2.1. Accelerometry

Raw accelerometer data is recorded as a signal of acceleration measured across axes; this can be analysed to detect regular patterns that represent movement [81]. This is most commonly measured across three axes in a tri-axis accelerometer: X, Y and Z (see Figure 4a). An accelerometer will measure linear acceleration across these three axes and by combining this with time, the signal can be used to quantify human movement and physical activity [81]. Specifically, movements including steps can be identified to track a step count whilst the general movement of the sensor can also be categorised into intensities of movement. The association between features extracted from an accelerometer with several health metrics has consistently been evidenced in research. Step count is one of the most commonly extracted features from accelerometer data (see Section 4.1.1) and has been linked to various health outcomes. A systematic review of over 13,000 adults identified that an increase in step count of 1000 daily steps above baseline was associated with a lower risk of all-cause mortality and cardiovascular disease (CVD) [82]. Similarly, there is a strong relationship between any recorded physical activity and a reduced risk of all-cause mortality [83]. Further, accumulated time spent in moderate to vigorous physical activity (MVPA) is linked to significant reductions in mortality risk [84]. This suggests that the collection of these variables prior to surgery may have a strong association with a patient’s health and could be useful for predicting complications.

The majority of research in the preoperative period utilises WS from commercial providers that have their own internal proprietary systems. As a result, researchers rarely analyse raw accelerometer data in the preoperative period. Multiple projects utilised a Fitbit wearable from which they could extract step count and energy expenditure [65,70,71,74,85]. Similarly, multiple studies utilised Garmin-derived metrics including step count, distance travelled and energy expenditure [40,76,86]. Data generated from a combination of mobile devices and WS that had a common platform (Apple Health; Achievement) extracted similar variables [87,88]. Particularly amongst large-scale research with many participants, these pre-calculated features were utilised for analysis rather than raw data [88,89]. Some research reported utilising pedometers as their WS to collect patient data; these pedometers all included accelerometers rather than a traditional step counter allowing them to represent the intensity of movement [90].

Although uncommon, some researchers did utilise the raw accelerometer data from the devices. Three studies utilised raw accelerometer data to stratify the signals into different intensities of activity [64,72,91]. This allows researchers to quantify the intensity of movement and compare time spent in sedentary behaviour against higher intensities including MVPA. Figure 3b presents a breakdown of the location at which sensors were worn. The majority of all sensor devices, and therefore accelerometer sensors, were worn on the wrist in commercial devices (Fitbit/Garmin) [71,76,80,86]. However, many of the research papers that only included an accelerometer sensor were worn on the participant’s hip, see Figure 3b [64,77,78,79].

**Figure 4 sensors-24-00482-f004:**
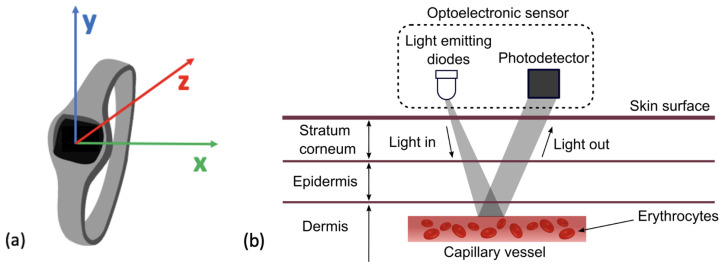
(**a**) Figure to show reference axes in a Tri−axes accelerometer. Presents the axes along which acceleration of movement can be measured across x, y and z. (**b**) The mechanism for HR detection in a PPG sensor by reflection. The LED can be seen emitting light which is reflected and then detected by the photo-detector and converted into a HR signal. This figure was taken from Moraes et al. (2018) with no changes made, Creative Commons Attribution International 4.0 License [92,93].

### 2.2. Photoplethysmography

Photoplethysmography (PPG), sometimes referenced as optical heart rate monitoring, is a sensor that can estimate an individual’s heart rate (HR). It utilises an optical emitter to give out light emitting diodes (LED) onto the skin that is attenuated from the pulse in the artery [94]. The reflection of the LED is captured by a photo-diode. The digital signal processor located in the device then translates this into heart rate data, as seen in Figure 4b. HR is one of the most commonly measured vital signs across medicine. It holds significant prognostic value for predicting general health and mortality. A lower HR is generally associated with a lower risk of cardiovascular mortality as well as a lower risk of all-cause mortality [95,96]. Research has also identified that a lower preoperative HR is associated with a lower risk of postoperative myocardial injury in patients undergoing non-cardiac surgery [97]. Therefore, cardiac assessments are common practice prior to non-cardiac surgery [98].

Previous research has investigated whether PPG produces comparable outputs to ECG. PPG sensors require both the LED and photo-detector to have contact with the surface of the skin and as a result, can be heavily impacted by movement or distance between the LED and skin resulting in optical noise [99]. A comparison of PPG to ECG signals can be seen in Figure 5a [100]. A large-scale study evidenced that HR estimates collected from PPG sensors correlate strongly with those from ECG signals [101]. Other research has concluded that at rest and at low HR levels, PPG has shown to have high accuracy but that this decreased with intensity of activity [102]. One publication reported the threshold for a reduction in accuracy of HR estimation to occur between 155–160 beats per minute [94]. This high threshold is notably above estimated HR maximum values for elderly populations indicating potential suitability for this population and for monitoring low-intensity exercise [103]. Most WS that include accelerometry at the wrist also include a PPG sensor [40,65,70,71,74,75,85,88,104]. One popular device that included a PPG sensor reported a data storage limit of 7 days and a charge limit of 10 days within the device [73]. PPG is a widely utilised technology in the preoperative period due to its convenience as a tool to measure HR. Wrist-based wearables have few requirements and are a practical tool for researchers as they require little input from users and have long periods of storage.

### 2.3. Electrocardiography

An ECG directly measures the electrical activity of the heart through electrodes placed upon the skin [105]. The electrodes measure electrical impulses from the heart that are then converted into an ECG graph. An ECG graph segment with annotations can be seen in Figure 5b. These annotations represent the detection of different stages in the cardiac cycle: the p-wave representing depolarization of the atria, the QRS complex representing the electrical impulse spreading to ventricular depolarization and the t-wave representing the re-polarization of the ventricles following contraction. The location and morphology of these annotations are used to extract several features including heart rate (HR) and heart rate variability (HRV) [106]. Using these features amongst others, an abnormal cardiac cycle can be identified from a signal and categorised. Heart disease is a leading cause of death worldwide making detection of cardiac abnormalities from the PQRST complex a key tool for preliminary diagnoses [107]. In clinical practice, a 12-lead ECG is common practice but this would be impractical for WS in a free-living environment. In this review, reduced lead ECG devices are considered wearable devices [108]. By analysing the ECG graph, HR can be calculated at any given time point. Additionally, using the RR interval (time portion between each R peak) other variables including heart rate variability (HRV) can be calculated.

An ECG sensor has several advantages as a prognostic tool compared to PPG sensors. The ECG is considered the ’gold standard’ tool for measuring HR; the accurate identification of a heartbeat allows the calculation of HRV from which further inferences about health can be made [109]. HRV has been highlighted as a tool that has promising value for predicting complications during and after surgery [110]. Additionally, an ECG allows for further detection of potential abnormalities including atrial fibrillation [111]; preoperative atrial fibrillation has shown to be predictive of complications in patients undergoing non-cardiac surgery [112,113].

One large cohort study used the Actiheart wearable ECG, which places two leads on the sternum from which three papers analysed the HR data [66,89,114]. Other research utilised an ECG ‘necklace’, which involved placing electrodes in the II lead configuration on the chest (see Figure 6c) whilst a further project included an ECG sensor that was integrated into a smart shirt (Hexoskin), see Figure 6b [115,116]. Although not utilising all 12-leads, wearable ECG devices have been shown to have high accuracy for heartbeat detection [105].

## 3. Pre-Processing of Signals

Pre-processing involves all changes to data that are made in order to prepare the data for analysis. Pre-processing can be the most vital stage in data processing and has a large impact on the inferences that can be made from a data set. Wearable data, even when collected in a controlled clinical environment, often requires heavy pre-processing due to the nature of the data. A wide range of pre-processing methods were implemented across the key papers and the most important techniques for each are outlined in Appendix A. There are two main challenges in WS data that pre-processing aims to overcome: missing data and noise.

### 3.1. Missing Data

Missing data is a frequently reported problem across research involving WS, particularly when using data from free-living environments [123]. Poor electrode placement, poor contact with skin or removal of device might lead to significant portions of poor quality or missing data. Often, the underlying reasons for periods of missing data are unknown. The prevalence of missing data in WS used in the preoperative period is outlined.

There is a wide variety of missing data reported across studies using wearable sensors in the preoperative period. Missing data was frequently reported at ranges of up to 25% from WS in this context [70,71,119]. One study reported that across all accumulative days of collected data, only 0.25% of days had complete HR data [71]. For the majority of research, the reporting of missing data refers to HR rate, rather than movement data (see Section 3.1.1). The reporting of missing data differed between research; some researchers report the overall percentages of data that were missing whilst others report the number of participants excluded due to missing data [40]. In both of these cases, research rarely goes into detail as to the causes of missing periods and how to categorise these. Missing data can generally be classified into three separate categories: missing completely at random (MCAR) where no systematic relationship is present between values that are missing and existing values; missing at random (MAR) where missing data is systematically related to existing data that has been observed but not unobserved data and missing not at random (MNAR) where missing data is systematically related to unobserved data [124]. Depending on the category of missing data that is assumed for the data, different methods may be better suited for minimising the potential bias that may be introduced [125].

Across research applicable to the preoperative period, three different strategies for handling missing periods of data were employed. As seen in Figure 7, missing periods of data were either deleted entirely, tolerated or imputed. These techniques have been previously identified in research using WS and are common solutions for missing data across fields [124,126]. To build on this, several techniques were identified in the present review that involve overlap between categories. To differentiate between data that has missing portions but is still usable versus data that should be deleted, an extraction threshold can be identified. Further, some research employs imputation on only short-term segments of data (see Section 3.1.3).

#### 3.1.1. Extraction Threshold

The extraction threshold identifies a point at which a subject’s data will be included in final analyses or is abandoned/processed further. This extraction threshold is usually only applied to wearable devices that measure HR in some format. For research that only utilised the pre-extracted step count, it is not possible to assess the exact volume of missing data. Step count data can appear as null values and still represent viable recordings indicating sedentary periods and so it is not always obvious to know whether this is as a result of non-wear, device malfunction or sedentary behaviour [127]. This is particularly true when step-count is only reported at the daily level [76,79,80].

These extraction thresholds differ widely between research. One study set a daily yield extraction threshold at a minimum of 8 h of collected data for that day to be included in analysis [71]. Other research set their extraction threshold at 10%, defining that any day with data of a daily yield above 10% would be included for analysis [70]. These studies indicate that the daily extraction threshold can be set at a relatively low value to allow for a high level of missingness in data and prevent this data from being abandoned. Research that used large data sets could set their extraction thresholds at a higher level; one study with over 80,000 participants only selected participants that had a minimum daily yield of 20 h [66]. However, this study did not report what percentage of participants had to be excluded as a result. Large data sets may have more flexibility in their extraction thresholds whilst a small research study may have to accept a higher level of missingness in order to prevent excluding a large portion of their sample.

A total yield extraction threshold can also be applied to the number of days in the monitoring period that have data [119]. This can be employed by only including participants that have above *x* number of days of data, with *x* indicating the threshold. The employment of a daily extraction threshold (i.e., 8 h of data) versus an extraction threshold for total data yield (i.e., 3 days of data needed) will depend on whether the data are subdivided into daily segments or kept as a total per participant.

#### 3.1.2. Selecting an Extraction Threshold

Extraction thresholds should not be randomly selected. To investigate the influence of the extraction threshold on the predictive performance of analysis, one study varied their extraction threshold from 1 to 10 h and identified that between 8 and 10 h achieves the best performance [71]. This highlights the importance of identifying an optimal extraction threshold. Setting a high threshold for inclusion will result in less data available for analysis; a low threshold has the potential to allow data from days with large missing periods into analysis. If this is the case then the underlying reasons for missing periods should be assessed to prevent bias in the data (see Section 3.1).

Aside from reported extraction thresholds within their own data sets, very little research has focused on quantifying the volume of data that is needed to obtain reliable preoperative baseline measurements. It has been reported that when using a PPG sensor combined with an accelerometer, a minimum of three days of monitoring should be completed; however, an extraction threshold within each day was not specified and whether the location and type of sensors have an impact on this threshold was not discussed [128]. An appropriate extraction threshold is a useful tool for selecting data for analysis but does not provide a solution to missing periods. To overcome the missing periods of data that remain, several imputation methods can be employed.

#### 3.1.3. Imputation

Data imputation in WS data is a complex process. Individuals will often have varying levels of missingness between them due to compliance with wearing the device. Further, there may be missing portions caused by technical issues in a device. Therefore, imputation techniques in WS data should generalise to both the participants’ behaviour and the device’s patterns [129]. Often in research collecting data in the preoperative period, data that was identified as missing was abandoned. Little effort is made to impute the missing values and why they are missing; this could reduce the size of the data sample and in some cases may introduce bias [130]. The imputation techniques that were employed in the research are outlined in the sections below.

The simplest method to replace missing data points in HR signals is with the mean HR values of activities at waking periods [88]. However, if the mechanism for missing portions is known to occur during sedentary periods or periods of vigorous activity then this may lead to under or over-estimations of daily HR. One particular study substituted missing HR values for HR recorded during a hospital visit [73]. This technique was not common across research, likely due to inconsistencies that may be present between free-living data and data collected in-clinic (see Section 1.2). For the studies that utilised the temporal aspects of data to employ imputation, they both did this using a two-layered pipeline [70,71].

The k-nearest neighbours (KNN) technique was shown to be a common method to impute missing HR values [70,71]. KNN has previously been implemented as a technique to address missing data in a range of applications [131]. The KNN algorithm is implemented as a ’sliding window’ that allows missing HR data to be calculated from a combination of recent step count and HR data [70]. This method is rationalised by explaining that imputation is useful for short-term missing segments where previous values of step count and HR have a high correlation with future values. One technique employed a k-nearest neighbours (KNN) algorithm for an entire day of data if the daily yield for the relevant day was above 10% [70]. A different study utilised the same technique but for all portions of missing data that were shorter than 10 min long, regardless of daily yield. If the segments of missing data were less than 10 min, a KNN sliding window (length of 5) utilised recent HR and step counts to predict HR values [71], see Figure 8. Using this method, the feature vector is imputed to the KNN algorithm where *hr_t_* and *step_t_* represent HR and step data at time *t*.

To assess the performance of various imputation techniques, several metrics can be employed. Root Mean Square Error (RMSE) is a commonly used metric for assessing imputation in signals from WS [129,132]. RMSE measures the average magnitude of the errors of the imputed values. It is particularly useful for assessing imputation because it penalises larger errors more heavily making it especially sensitive to values that are big outliers from the predicted values. Mean Absolute Error (MAE) provides an alternative to the RMSE in that it is not sensitive to outliers. It presents the absolute differences between the imputed and original values and is focused on the overall accuracy of the algorithm. It may be useful to compare the outcomes of the MAE with the RMSE and the two have previously been used in combination [129]. For the MAE and RMSE, lower values indicate better performance. However, in order to assess imputation techniques using these metrics, ground truth values are needed. One method to do this is to add missing periods of data into a signal and perform imputation on these fabricated missing periods to compare outputs against original values as ground truths. It should be noted that this has been very rarely implemented in research across this field.

#### 3.1.4. Feature Level Imputation

In instances where data has been abandoned due to significant periods of missing data, this can be imputed by employing feature-level imputation techniques. After abandoning days with a daily yield below the extraction threshold, a feature-level imputation technique can be employed to compute the features that represent the days with high portions of missing periods. In one case, researchers again utilise the KNN method to impute statistical and semantic features based on the neighbouring features that are available for that participant, rather than attempting to impute the missing values in the signal [70]. A further technique to deal with missing data that fell below the daily yield of 8 h was to employ imputation on high-level features that have been calculated from daily features [71]. Further application of Detrended Fluctuation Analysis was used to reduce the incomplete data.

It is important to note that research only employed imputation for HR data, this was not performed for other features extracted from accelerometer data. For research where HR signals are collected alongside step count data, the proportion of missing step count data can be extrapolated from the time periods with missing data points of the HR signal. Step count data is reported as being significantly less correlated and so less predictable than HR data [71]. Instead, these data were normalised by dividing the step count by the daily yield so as to prevent the step count from being drastically increased just for those patients with more data accumulated.

### 3.2. Noise

Aside from missing data in the signal, noise can also prevent meaningful features from being extracted. Accelerometry data can be plagued by white noise, altered by human motion or vibration whilst both ECG and PPG signals can be corrupted by motion artefact, baseline wonder and electromyography (EMG) noise [133,134]. Few papers utilised raw signal data (see Section 2) meaning filtering of signals was not commonly reported. When techniques were employed to filter noise, this was performed on the accelerometer and HR data separately.

In the present review, many WS devices only report pre-calculated HR values from internal algorithms meaning raw cardiac signals were rarely processed. In certain cases, some removal of noise from signals was completed. Previously, Gaussian process robust regression has shown to be successful for noisy HR data and was implemented by one study to utilise prior knowledge of the HR data to reduce noise [66,135]. In comparison, a simpler method to limit the noise in HR data was to average the HR extracted from R-R intervals over a set time period, this varied from between 15 s to 15 min [114,115]. When employing this technique, all inaccurate HR values were identified and removed from the data where consecutive HR values varied by more than 20% [115]. Cardiac signals were very rarely passed through a low-pass filter; however, one study resampled the HR to 1 Hz before passing HR through a 0.01 Hz low-pass filter to remove high-frequencies affected by non-linearities introduced from circulatory distortions [116].

For accelerometry, to convert the raw signal data into magnitude of acceleration the Euclidean norm minus one and high-passed filtered vector magnitude were used [66]. Altini et al. (2016) employed a different filter technique where a low-pass filter (1Hz) was used to isolate the static component in the signal due to gravity and a band-pass filter (0.1 Hz, 10 Hz) was used to isolate dynamic components due to body noise [115]. As mentioned above, Beltrame et al. (2017) used a similar low-pass filter at 0.01 Hz for accelerometer as well as HR data [116]. The only implementation of a fast Fourier transformation (FFT) was to integrate the frequency in accelerometer data between 1 Hz and 10 Hz [91]. One particular paper reports a method for outlier detection within data by removing values that are greater than 3 standard deviations from the mean [104]. Building on the techniques commonly employed on HR data by averaging values over a short period, research employing a pedometer categorised each accelerometer period of 10 s into either lying, stationary or active periods [91].

For research using large-scale cohort data, after normalising their data through standard scaling with unit variance, researchers applied Principle Component Analysis (PCA) onto the original training data set that retained the components that explained 99.9% of variance [89]. To prevent any information leakage across the data sets, the fitted PCA scaler was applied individually to the test set. In a different project utilising the same large dataset, researchers attempted to reduce the noise that is seen in the labelling of HR data from the ECG wearable using deep learning [114]. The authors propose UDAMA (Unsupervised Domain Adaptation and Multi-discriminator Adversarial) training network [66]. However, these techniques utilising deep learning will require a large pool of data and may not be suitable for smaller single-centre studies. In order to reduce the noise that is present in daily features, one research study utilised singular spectrum analysis to further extract high-level features from daily features [71]. This allows trends to be extracted from the noisy data with missing portions by computing the mean, variance and slope from each time series of daily features.

### 3.3. Encoding Time

The temporal aspect of the HR and accelerometer data can provide useful insights but is infrequently used in analysis. To investigate the temporal aspect or reduce the bias that can be related to the time of recordings, it is possible to encode timestamps from sensors [66,89]. This is performed to help capture any periodic nature of certain behaviours, particularly those that may exhibit daily or monthly habits. By incorporating these cyclical components the model can better understand the cyclical nature of time. This was completed by encoding either the month of the year or hour of the day as (x, y) co-ordinates on a circle using Equations (1) and (2) where *t* represents the temporal aspect.
(1)$$T_{f_{1}}=\sin(\frac{2*\pi *t}{max(t)})$$
(2)$$T_{f_{2}}=\cos(\frac{2*\pi *t}{max(t)})$$

This is to ensure that when including time in analyses neighbouring timestamps are always consistent; December is only one month from January and 23:59 is only one minute from 00:00. This is particularly important for large data sets that are likely to be collecting data over an extended time period.

## 4. Feature Extraction

Feature extraction is an important step in signal processing to convert the signal data into numerical features that can be processed in a model [136]. It can also be useful to reduce the dimensions of the data when a large amount of data is collected [137]. Most research performed feature extraction from each signal separately but where possible features were extracted from multiple signals. The resulting features are outlined, and a full breakdown of the relevant features extracted from selected papers can be found in Appendix A.

### 4.1. Features Extracted from Accelerometer Signals

#### 4.1.1. Step Count

Step count is a frequently reported feature that is used in WS research. Extracting step count from raw accelerometer signals involves implementing an algorithm to detect a pattern in the data, the choice of algorithm might depend on the computational complexity and resources that are available. Common algorithms include the peak detection algorithm, which identifies the local maxima and minima (see Figure 9) or simpler thresholding algorithms that identify optimal thresholds to detect steps [138,139]. Machine learning techniques also have shown good application but may require labelled data to train models, which are not often available [140,141].

Much of the research that utilised commercial devices employed internal proprietary resulting in pre-calculated features. If the raw signal is automatically converted into step count then any further features relating to the intensity of movement are not able to be extracted. Frequently, steps are reported as an average measure across all days meaning any useful temporal aspect of this data will be lost [76,77,80,86].

Patients can be further classified into groups based on their average step counts to then produce features labelling individuals as active/inactive. Often research used pre-determined thresholds to classify participants into these groups. Threshold values range from 2500 to 5000 daily steps and there is no consensus on the threshold that should identify an individual as ‘active’ [73,77,80]. Another technique is to define these thresholds based on the variance of step count within the research cohort so that the split of participants in each group is even [79]. It is not conclusive which of these methods to stratify patients into activity groups is most conclusive but this should be relevant to the research cohort and context of the research [143].

#### 4.1.2. Movement Intensity

Where raw signal data is available, the intensity of movement can be calculated from the acceleration and as a result, the time spent in sedentary, moderate to vigorous and vigorous activity can be reported [64,72]. This is calculated from time spent at activity counts above a specified threshold. Activity counts are calculated from ActiGraph’s proprietary algorithm; this algorithm has been widely used across research employing accelerometers and has been published as open access software [144]. However, the selection of a cutoff point for different intensities of physical activity results in significant differences in total MVPA between research and there is no standardised cutoff points [145]. Therefore, care should be taken when selecting a cutoff that is suitable for the research population.

#### 4.1.3. Distance Covered

By utilising the features that can be extracted directly from the accelerometer signal, further features can be inferred. The distance that is covered by a participant can be calculated from a combination of the number of steps times that are taken in a day multiplied by the stride length of the participant [65,78]. It must be noted that stride length should be adjusted for using further participant information including sex and height information.

### 4.2. Features Extracted from Cardiac Signals

#### 4.2.1. Heart Beat Detection

HR signals are complex and vary depending on sensor modality. As outlined previously, PPG and ECG signals are the most commonly collected cardiac signals that are used. To calculate HR from these signals, a process of data cleaning and beat detection is employed. The Pan-Tompkins algorithm (PT) is the most widely used beat detection algorithm which has been shown to be capable of detecting the location of a QRS complex in the signal across both clean and noisy data [146]. The PT algorithm employs a band-pass filter to isolate the relevant frequency before using a combination of thresholding and dynamic adjustment to identify the R-peaks. Other popular beat detection algorithms include wavelet-transform-based methods that analyse wavelet coefficients or simple algorithms that look for peaks in local maxima [147].

#### 4.2.2. Heart Rate

By using the locations for the QRS complex that have been detected, see Figure 5b, HR at any given time can be calculated. As previously mentioned, the majority of research that is applicable to the preoperative period utilises commercial devices where HR values are often pre-calculated from the detected beats using an internal proprietary algorithm in the WS. As a result, it is rare that research in the preoperative period has to detect the location of a QRS complex. Instead, HR values are given at a varying frequency. The update period of the heart rate signal will dictate how regularly the heart rate is updated, commonly this is updated every beat. As a result, HR signals from WS are often extended signals that require further processing to extract meaningful features.

#### 4.2.3. Resting HR

Resting HR (RHR) is a term that does not have a consistent definition but generally refers to the HR of an individual when they are inactive [148]. RHR is widely considered an important bio-marker of physical health and has been shown to be associated with both mortality and morbidity after non-cardiac surgery [97,149,150]. Although RHR is widely accepted as an important biomarker, there are also no set guidelines in medical literature for calculating RHR. Recent literature has suggested that when employing WS in research, a minimum four-minute rest time is required for a reliable RHR measurement [148]. From these values, a resting HR can be calculated. One research project reported calculating resting HR over a 24-h period but made a distinction between resting HR recorded during the night [73].

#### 4.2.4. HR Changes

From the 24-h period, time spent in different HR zones can also be extracted as an indicator of activity throughout the daily period [73]. Other research utilised HR signals to create a new variable by assessing the difference between a current HR value and a previous value at a 1 s lag to represent ’dynamic changes’ in HR and cardiac activity [116]. Other research also employed a two-level feature extraction pipeline where first-order statistical features like skewness and kurtosis of the HR are extracted and high-level features are then taken from the daily level features including the slope, mean and variance [71].

#### 4.2.5. HR Variability

HR variability (HRV) is a metric that is calculated from the variations in intervals between detected heartbeats. It is an accepted tool for measuring the function of the autonomic nervous system, a vital factor in cardiovascular health [151]. Recent research has confirmed that preoperative HRV can be a useful predictor of postoperative outcomes [110]. However, in the preoperative setting, HRV has traditionally been calculated in a clinical setting using only ECG signals rather than employing WS [152].

Recent WS research has shown promising efficacy in calculating HRV from noisy signals in both PPG and ECG signals [153,154]. This highlights the potential for HRV to be used as a preoperative tool calculated from WS. One study did calculate HRV from participants’ ECG data by differencing the second-shortest and the second-longest inter-beat intervals [66]. There are variations in how HRV is calculated depending on whether they are time or frequency domain features. A common time domain frequency measure is the square root of the mean of the sum of the square of differences between NN intervals (RMSSD); both Garmin and Fitbit devices employ RMSSD to measure HRV in their commercial devices [155,156]. Although these devices may be able to report HRV, it is not always available to be extracted for use in research; one particular study reported that HRV, although measured by the sensor, was not able to be extracted from the device for use in research [73]. RMSSD is presented in Equation (Equation 3) where NN_i_ is the length of time of the *i*th interval and N is the total number of NN intervals in the dataset.
(3)$$RMSSD=\sqrt{\frac{\sum_{i=1}^{N-1}{\left(NN_{i}-NN_{i+1}\right)}^{2}}{N-1}}$$

Frequency domain features are calculated using estimation of power spectral density and include a range of features such as low frequency (lf) and high frequency (hf) [157]. Further non-linear properties of HRV can be analysed and extracted using Poincare Plot and Sample Entropy; however, these are less frequently calculated.

### 4.3. Multi-Modal Sensor Feature Extraction

#### 4.3.1. HR Recovery

Where possible, a combination of signals can be used to form features. HR recovery was calculated by measuring the decrease in HR one minute after exercise cessation which can be identified from movement data in accelerometer signals [65]. HR recovery has previously been shown to be predictive of cardiovascular health and it would be unknown whether the activity has completely stopped without the incorporation of accelerometer data [158].

#### 4.3.2. Respiration Rate

Respiration rate (RR), also known as breathing rate, is a parameter that is commonly collected from WS but has been infrequently used in preoperative research. Across multiple clinical settings, RR has been identified as a valuable bio-marker of health status and has even been suggested as a predictor of in-hospital mortality [159]. On commercial devices, the most common implementation is to calculate breathing rate using the HRV data from PPG sensors [155,156]. Estimating respiration rates that are calculated from PPG sensor data is a well-established technique but has come under critique due to its sensitivity to noise [160]. There are a wide range of mathematical techniques that are proposed to estimate RR from PPG and ECG signals but the majority follow a general pathway. This involves extracting portions of the original signal that are dominated by respiratory modulation before fusing these signals to output one signal from which RR can be estimated [159]. A further method is to incorporate respiratory bands into a wearable device; Beltrame and colleagues (2017) [116] used RR calculated from respiratory bands to successfully predict oxygen uptake for participants.

When worn on the chest or torso, an accelerometer is another non-invasive method for estimating RR. RR is calculated by measuring movement and angular changes in the chest [161]. Some research has suggested that wrist-worn acceleration devices can also be used to accurately predict RR, but that this is only possible during periods of non-physical activity [162]. A further paper proposed a multi-sensor solution that employs sensors at several locations to predict respiratory rate [163]. There is no threshold for the acceptability of these devices when compared against gold-standard measurements (breaths counted by a specialist) meaning it is unclear which devices are best, although one study proposed a difference of under 2 breaths per minute to be acceptable [164]. Although this research suggests that there are multiple options for estimating RR from devices, research that has application to the preoperative setting has very rarely employed this feature in its analyses, similar to HRV.

## 5. Data Analysis

Research extracting data from individuals’ behaviour and vital signs from WS use a large variety of computational techniques, this can be seen in Figure 10. The chosen models range from simple statistical analyses to complex models with high computational requirements. The variation in models may be explained by several factors including the outcome that is being predicted, the size of the data set and the nature of the features that are extracted from signals. The models employed are separated by their complexity in the following sections, a breakdown of the methods employed within key papers can be found under Appendix A:

### 5.1. Feature Selection

Due to the vast range of features that can accumulate from free-living WS data, feature selection is the first stage in model development. This is important to assess the relevance and the effectiveness of the extracted features from the WS in relation to predicting outcomes. The most common feature selection method was to perform univariate analyses on features. Using this method, the relationship between each extracted feature and the target outcome is assessed. This evaluates each feature independently to determine its significance to the outcome variable. The features that are then found to have a strong relationship with the outcome are selected for further modelling. Pearson’s correlation was tested between all physical activity features and cardio-respiratory measurements, those variables with a *p*-value below 0.05 were excluded [87]. Within the remaining features, those with a covariance of over 0.7 were compared and the feature with the highest Pearson correlation would be included in the final multivariate model. A similar technique was employed but measured the correlation with wearable data and postoperative complications [72].

For studies that separated participants into groups based on thresholds in their WS data, chi-squared tests were used to test for differences in categorical variables and Fishers, Kruskal–Wallis or a Wilcoxon sum-ranked tests were used to test for differences in continuous variables [77,79]. Variables with a *p*-value that was below the set significance level for inclusion in final models were then included in multivariate analysis; the significance level for inclusion varied (*p* < 0.05–*p* < 0.1). A similar technique was employed by instead grouping patients by their postoperative outcomes (re-admitted patients versus non-re-admitted patients) [85]. Where multivariate regression models were employed, all preoperative variables were included in the initial model. A backward-stepwise selection using the Akaike Information Criterion (AIC) was then used to assess each variable’s importance in the model [80,165]. Feature selection is an important stage in model development as it can lead to enhanced model performance, reduced data dimensions or enhanced interpretation. This reduction in dimensions may also help prevent over-fitting of the model to the training data.

### 5.2. Univariate Analysis

The earliest study to investigate the relationships between preoperative physical activity (PA) levels and postoperative complications using a WS stratified patients based on their preoperative PA levels [64]. They performed independent *t*-tests within PA groups between those who did or did not develop a complication. More recent research employs similar univariate analysis between groups but utilises a Chi-squared test for categorical variables or a Wilcoxon sum-rank test for continuous variables [65]. This is the simplest method to assess if there are significant differences between groups and therefore was most commonly employed (see Figure 10). Although useful in a preoperative setting to stratify patients into high and low risk based on identified thresholds, the models themselves do not allow any predictions to be made regarding patients’ risk of complication.

### 5.3. Correlation Analysis

Correlation analyses were a further common tool to investigate WS data. Greco et al. (2023) performed correlation analysis on preoperative daily step counts and the outputs of a 6MWT [65]. They showed that a strong correlation was present between the daily steps and clinical measurements taken preoperatively. Jones et al. (2021) also investigated the association between wearable variables and preoperative fitness measurements using correlation analysis but built on this using linear regression models [40]. The calculation of a correlation coefficient was used to investigate the relationship between preoperative step count and postoperative complications [76]. These models on their own do not allow predictions to be made in a clinical setting but are often useful as a preliminary tool to test for the presence of a relationship.

### 5.4. Machine Learning (ML)

Amongst papers investigating the link between preoperative WS data and outcomes, ML techniques are widely implemented. ML methods are well suited to non-linear, large-scale data. However, because the majority of research in the preoperative period has a relatively small number of participants and pre-processed data, not all ML models will be useful. Machine Learning methods are sub-categorised below:

#### 5.4.1. Logistic Regression

Logistic regression was the most commonly implemented technique used in the clinical setting. Patient groups were either split by their preoperative activity levels or by their postoperative outcomes. Multiple research investigated the ability of preoperative PA to predict postoperative readmission as a binary outcome [79,85,86]. Internal validation techniques were reported by Rossi et al. (2021) using four-fold patient cross-validation and regularisation in the model to prevent the model learning noise and overfitting [86]. In comparison, another study employed leave-one-patient-out cross-validation (LOPO) to allow a change in the distribution of the data and prevent the model from overfitting to the training data [75]. This technique is a process where one patient is left out as the validation set and the validation is performed k-number of times, where k is the number of patients [166]. Other papers used a similar technique but where multiple outcomes were recorded, multiple models were used. To assess model performance in the case of multivariate logistic regression the C-index (Harrell’s concordance index) was reported to be a useful measure [80].

#### 5.4.2. Multivariate Regression

Rather than a logistic regression, multiple studies used multivariate regression techniques to predict continuous outcomes. Bille et al. (2020) used multivariate regression to predict the number of absolute complications per patient [79]. Mylius et al. (2021) used this to predict the time to functional recovery as an overall measure of recovery alongside the odds ratio of the occurrence of complications [72]. Novoa et al. (2011) used two separate linear regression models with bootstrap robust estimation (1000 iterations) of the standard error of regression coefficients. The first model was tested with mean daily distance walked as the independent variable whilst the second model incorporated distance travelled [78].

#### 5.4.3. Ensemble Models

Ensemble models represent a development in Machine Learning where a combination of models are used to obtain better predictive performance [167]. Random Forest models were shown to significantly outperform the LASSO (Least Absolute Shrinkage and Selection Operator) models when predicting laboratory-based measurements [104,168]. Random Forest models use a combination of decision trees for solving a problem by aggregating them together to perform as a group rather than alone [169]. This may be related to the structure of the data and indicates that these models are superior when utilising non-linear data. When comparing models, there is no consensus on a particular ML model that is most suited to free-living wearable data and is likely dependent on a large variety of factors.

#### 5.4.4. Support Vector Machines (SVM)

Ensemble models were not always shown to be superior to other individual ML methods. One study implemented a range of models: random forest (RF), k-nearest neighbours, XGBoost and SVM [71]. In this case, the SVM outperformed the remaining models by a larger margin for predicting postoperative complications. However, in a similar research case where SVM and RF models were compared against Gradient Boosted Trees (GBT), GBT models were found to have the highest performance [70]. In this case, the LOPO internal validation technique was implemented which may have influenced the performance of prediction models.

Other applications of the SVM technique were as a precursor to the final model. This was used to identify the portions in the extended HR data that would be most suitable for predictions. Data collected from simulated activities in the lab were used to recognise activities that are then completed in free-living environments [170]. SVM was used for pattern recognition in the free-living data. Acceleration was particularly useful for classifying high-intensity activities. SVM was used to find the optimal discriminatory boundary between activity clusters.

The range of ML models implemented across research displays how there are several options that may be suitable when predicting outcomes from WS data. However, there is no consensus on which ML techniques are best suited to this data type; particularly in research comparing the application of several ML models there have been confounding conclusions [70,71]. This is likely due to the range of factors that will influence the performance of these models, including the sections discussed previously.

### 5.5. Deep Learning

No deep learning (DL) techniques employing neural networks were utilised in the research investigating the associations between preoperative wearable sensor data and postoperative outcomes. However, there were multiple studies that utilised DL when predicting clinical fitness measurements. These were employed when analysing large cohort data sets [66,88,89].

#### 5.5.1. Predicting HR Response

One paper employed an attentional convolutional neural network to identify and learn the signatures of different cardiovascular responses to data collected [88]. Specifically, a HR auto-encoder was trained to produce the given HR from physical activity and sleep stages. To learn the personalised cardiovascular response functions from the wearable data, the HR encoder was trained on physical activity and sleep stages. The encoder stage learns the signature of an individual from their HR responses to exercising or movement and the decoder employs this learned signature to predict HR based on movement.

Further research uses DL techniques in a similar format to the aforementioned research but instead proposes a general-purpose model that does not require a historical input of one month [66]. This paper proposes the ’Step2Heart’ receiving high-dimensional activity inputs to predict HR response which similarly uses the accelerometer data to predict HR. Stacked CNN and RNN layers are combined where the CNN learns spacial features and the RNN learns temporal features of the data. Aside from predicting a future HR response, this model has also shown to have further clinical value in the preoperative period in that it can be utilised to predict VO_2_ max values.

#### 5.5.2. Predicting VO_2_ Max

Using a large WS data set, a deep neural network was able to successfully predict VO_2_ Max measurements from cardio-respiratory testing. The network employed two feed-forward layers with 128 units that are densely connected [89]. Batch normalisation and dropout were included to help prevent over-fitting. The final layer in the model is a single unit layer; the model was trained using a pre-trained optimiser (Adam optimiser) to minimise the mean squared error [89]. The model was able to outperform other models and was also able to predict future changes to VO_2_ max recordings.

Similar findings were found when deep learning was used to both clean noisy data and to predict cardio-respiratory fitness as a real value [114]. Due to the large nature of the data when WS are worn for an extended period, features were extracted from the raw signals and represented as feature vectors. The deep neural network trained on these inputs was able to achieve high performance when compared to the ground truth values from cardio-respiratory testing [114].

## 6. Future Challenges and Opportunities

This literature review has identified research utilising WS that is applicable to the preoperative period. Several findings were made that relate to the four subsections of the review. These are outlined below and described in the context of future challenges for research.

### 6.1. Comparison of Sensor Modalities

As is apparent from Figure 4a, the use of accelerometers is widespread across WS in research as a health assessment tool; a smaller subset of research employed an additional sensor to record cardiac signals (ECG or PPG sensors). Although less data is recorded from participants, there are several benefits associated with employing only an accelerometer in the wearable sensor. An accelerometer device will rarely require input from the user and advantages include simplicity and decreased power consumption. The RT3 accelerometer used by Feeney et al., (2011) collected data for up to 21 days without input whereas the ECG necklace device used by Altini et el., (2016) required daily charging [64,115,171]. Reducing periods of non-wear for charging throughout the study may reduce the likelihood of missing data. Further, accelerometer devices are not reliant on skin contact like PPG and ECG sensors which might further decrease periods of missing data (see Figure 4b). In applications where cardiac data may not be necessary, accelerometer sensors are simple, unobtrusive devices that collect valuable recordings regarding participants’ activity levels.

However, when accelerometer signals are combined with signals from an ECG or PPG it allows for more comprehensive health monitoring. Cardiac data from PPG/ECG signals has consistently shown to have strong associations with CVD and poor clinical outcomes; this allows for interpretation of cardiac health that is not possible from accelerometer devices alone [172]. The incorporation of cardiac signals also allows for the detection of specific health conditions including arrhythmia’s [172]. Further, multi-modal signals can improve the evaluation of free-living data. In research directly comparing the application of sensor modalities, it was found that activity recognition had superior accuracy when combining an ECG recording with accelerometer data [173]. Other benefits that might arise when combining the two signals include an improved assessment of signal quality. Combining a PPG with accelerometer signals has previously been shown to allow for signal quality analysis that is not possible with each signal independently [174]. By including cardiac signals alongside movement data, a combination of sensor modalities can allow for a more holistic assessment of patient health. This includes the extraction of further features that require multi-model signals (see Section 4.3) and an in-depth analysis of an individual’s cardiac health whilst also optimising the classification of activities. Although this adds value, it does come with the expense of added complexity both when collecting the data itself through the limitations associated with multi-modal sensors and the analysis of added complex signals. The attributes of these devices should be compared and sensor modality should be selected based on the context of the research.

In the research included in this review, patients were commonly recruited once surgical treatment had been scheduled or at the preoperative assessment clinic. As a result, any WS data from patients was usually collected in the immediate weeks preceding surgical treatment (see Section 1.4). There was no variation in the data collection period between sensor modalities; however, the preoperative period is by definition a broad label for any preoperative assessment (see Figure 1). The varying characteristics of sensor modalities outlined above indicate that simple devices with large data storage and battery capabilities may lend themselves to data collection periods extending over a longer period whilst more complex devices that incorporate multiple signals may be useful at shorter intervals to assess vital signs. Therefore, different sensor modalities may be suitable at different stages prior to surgical treatment but there is no existing literature discussing this.

A further area of interest within the field of preoperative wearable sensors is the development of new sensor modalities that can be relevant to the field. Many of the issues relating to the causes of missing data may have the potential to be overcome through the use of new materials. For example, poor electrode positioning of an ECG lead may lead to missing periods due to excessive movement. A sub-class of flexible sensors known as Piezo-Resistive sensors have shown to have a linear response to bending and elongation and as a result, are highly sensitive [175]. This means they have shown to be capable of measuring precise human movements like breathing and exercising whilst also being able to measure a pulse [176]. The development of these materials as sensors that can collect vital signs from patients whilst allowing significant movement may provide a better and more comfortable alternative than current solutions.

### 6.2. Missing Data Periods

When collecting WS data, there are often significant periods of missing data. This was seen in the frequency of studies reporting missing periods in signals (Section 3.1) and several solutions to this are outlined. Pre-processing of data is a key component of the analysis timeline and missing data is not always well addressed. Across the studies, there was no set protocol for handling missing data. The underlying reasons for missing data periods are rarely thoroughly investigated and this highlights a significant problem in the literature. This highlights a challenge in the field that should be addressed; a protocol for assessing the reasons for missing data might help research better address the missing periods with more appropriate solutions. Within the studies that do address missing data, there is some mention of the underlying reasons. This includes non-wear time or poor electrode positioning. One research paper refers to the frequent need to replace electrodes and this highlights an issue that may lead to an increase in missing periods of signals [115]. In order to overcome this, future research should compare the missing periods that are associated with different sensor modalities and whether specific techniques can be employed to address this.

Extraction thresholds are often used to identify when there is a suitable volume of data collected either over the course of each day in the monitoring period or over the course of the entire recording. Although some research justifies the selection of an extraction threshold through testing, this is not always the case and an arbitrary unit is selected. Some research reports abandoning data whilst others report using imputation techniques. Future research should focus on identifying an appropriate protocol for overcoming missing data in WS research as there are significant variations across the field.

### 6.3. Raw Signal Data

The majority of research that is applicable to the preoperative setting utilises commercially available sensors that employ their own internal proprietary algorithms, a common example being the Fitbit Inspire [65,71,73,85]. Researchers are provided with pre-extracted features; for example, heart rate and step count, where no pro-processing is required. This can improve access to research as data from these sensors is computationally simpler to work with; however, this limits researchers in the agency that they have for pre-processing and feature extraction. In research predicting VO_2_ Max, step-count and floors-climbed were extracted by an internal algorithm from a wrist-worn sensor [40]. Other research has shown that acceleration-derived Metabolic equivalent of Tast (METs) and raw acceleration alongside step count data has been shown to be predictive of cardio-respiratory fitness [89]. Further, a variety of pre-processing techniques can be applied to accelerometer signals (see Section 3.2) and when using proprietary algorithms, research may be limited in how it can filter signals. Proprietary algorithms may limit researchers in the features they can use from the raw accelerometer signals.

Similar conclusions are particularly pertinent for cardiac signals from ECG or PPG sensors (see Section 4.2.2). Often HR is the only extracted feature from these signals and although HR is a useful measure with strong associations for health (see Section 2.2), there is a vast amount of information that can be further extracted from cardiac signals. HRV, calculated from the location of the QRS complex (see Section 4.2.5), has been a popular metric of health but is not able to be extracted from a HR signal. Research has suggested that not only ECG but PPG signals are adequate for estimating specific HRV features that are relevant for assessing patient deterioration [177]. Further, recent research has identified the potential for these cardiac signals from wearable sensors to identify patients at high risk of suffering cardiac abnormalities [178]. This highlights a large gap in useful data from both ECG and PPG signals that is not being utilised within research applicable to the preoperative setting. Although proprietary algorithms can simplify data access to wearable sensor data by removing the barriers of pre-processing data, it may in turn limit research capabilities by reducing the dimensions of the data and preventing extraction of features. Future research should investigate the added prognostic value that these underutilised features from raw signal data can bring to the predictive performance.

### 6.4. Predictive Models

There is a wide range of models being employed to investigate differences in outcomes. This ranges from models that compare characteristics across groups (active/inactive) to predictive models that predict the risk of complication/readmission/CPET. Machine learning models that utilise pre-processing and feature extraction techniques have been shown to be successful at predicting both postoperative outcomes and clinical fitness measurements. Due to the nature of data when combining HR with accelerometry, there are many different potential features for extraction. However, most feature extraction techniques in this review do not utilise the temporal aspect of the signals. Investigating how temporal data can be incorporated into feature extraction (e.g., HR recovery, see Section 4.2.4) and how this impacts the predictive performance of models should be researched further. Further, ML models have unique advantages that allow non-linear relationships to be analysed. All of these factors should be considered when selecting a model and where appropriate, multiple models should be compared. When using data from the preoperative period, there is no single ML model that clearly outperforms others. This indicates that future research should consider multiple models for predicting outcomes.

Deep learning methods were shown to be successful at predicting clinical fitness measurement research using large sample sizes. Although this suggests that DL methodologies may have application to wearable sensor data in this setting, there are several points to consider. Research utilising preoperative data requires approaching prospective patients to wear a sensor prior to undergoing treatment rather than utilising pre-existing data sets. As a result, preoperative research often has a low average sample size; DL models in this setting with low sample sizes will be disposed to over-fit the data [179]. Additionally, proprietary algorithms that limit the number of features that are extracted from signals may exacerbate this (see Section 6.4). A combination of these factors suggests that DL models may struggle to generalise these patterns effectively outside of the training data. Other considerations when implementing DL are the high computational requirements and access to this; the hardware required to integrate DL into predictive models is not accessible to all research groups. Whilst this is the case, shallow machine learning models have been shown to be suitable for predicting postoperative outcomes. Future work should investigate how to incorporate DL to large volumes of raw data from WS. Huge data sources are available when utilising raw data and therefore, deep learning techniques including Generative Adversarial networks may be suited to cleaning and pre-processing of data.

### 6.5. Considerations for Implementation

For research involving the collection of free-living data, much consideration should be given to the ethical implications that come with using this. Firstly, wearable sensors can collect a vast amount of highly personal data over a long period of time, referred to as Patient-Generated Health Data (PGHD) [180]. This results in a large stored set of sensitive data; ensuring that this is stored in a secure environment and is used in a method that will not impact negatively on participants is vital. Previous research has shown that understanding privacy concerns and legislation in WS data can have a significant impact on privacy concerns amongst patients and that this in turn may impact the self-disclosure of participant data. Rarely did the research in this review discuss the privacy and storage considerations for research but this was likely not a main research aim across the studies. Several regulatory frameworks exist that provide guidance on how to safely utilise individual data, including the General Data Protection Regulation [181]. One particular recent publication outlines recommendations for addressing privacy concerns regarding WS data and suggests that amplifying user agency over health data is a key measure to address this [181]. Given that this may be a determining factor in participants contributing data to research, future studies should consider the understanding that patients have over their data and any concerns that they might have.

A further point that is not particularly discussed across research is clinicians’ views regarding WS data. Although patient uptake is key to collecting data for research purposes, without clinician input and buy-in for the technology the benefits will be limited. Preliminary research suggests that clinicians value the benefits that WS can bring to the healthcare setting, particularly as a tool to provide feedback [182]. However, other qualitative research has identified an overarching concern regarding trust and lack of clarity over data control. Increased regulatory efforts have been discussed as a solution and this was highlighted as a priority for future work [182]. It is possible that this may become a component of future regulatory and clinical validation criteria. Across the research discussed in the review, there is limited reference to clinical validation of devices. Under current legislation, the FDA does not classify wearable sensors that ’maintain or encourage a healthy lifestyle’ as a device that requires approval and as a result, many of the devices referenced throughout the review do not require FDA approval [183]. Given the increasing evidence of the benefits that these devices can provide to the perioperative pathway, the regulatory frameworks and clinical validation process for patient data from WS may change. This may also impact the research requirements that are required to ensure that the findings from research in this field are valid. For example, much research in the field did not report on participant ethnicity, and when reported it was largely white participants (see Section 1.4; Appendix A). Unrepresentative study samples highlight a pitfall in the research field that should be addressed before findings can be generalised to the wider population.

Finally, if wearable sensors were to be implemented as a tool across the perioperative pathway, a cost-benefit analysis would be required. At the current time point, limited literature has evaluated the costs associated with WS in perioperative medicine, particularly at the preoperative stage. Given that much as the research in the review is regarded as feasibility research into the application of WS, it is likely difficult to estimate the savings in healthcare costs that are associated. Having said this, recent research has suggested that for patients with limited access to healthcare facilities following surgical treatment; for example, rural communities or those reluctant to return to facilities for repeat care, WS can provide an alternative that is cost-effective [184]. Other research has highlighted that WS can be a cost-effective tool to monitor patients across the perioperative pathway but mirrors the conclusions from this review that more research is required to conduct a cost-benefit analysis.

### 6.6. Conclusions

This review presents the recent literature relating to WS in the preoperative period. Several sensor modalities are discussed in detail alongside common pre-processing methods. The wide range of features that can be extracted are outlined alongside the models that most effectively utilise these features to analyse WS data. Several important points are highlighted in the final chapter alongside the gaps in the research field. Particularly, WS data is often not analysed in its raw form and this may limit the capabilities for pre-processing and feature extraction. From this review, several directions for future research are suggested including a strong focus on utilising raw signal data for analysis.

## Figures and Tables

**Figure 1 sensors-24-00482-f001:**
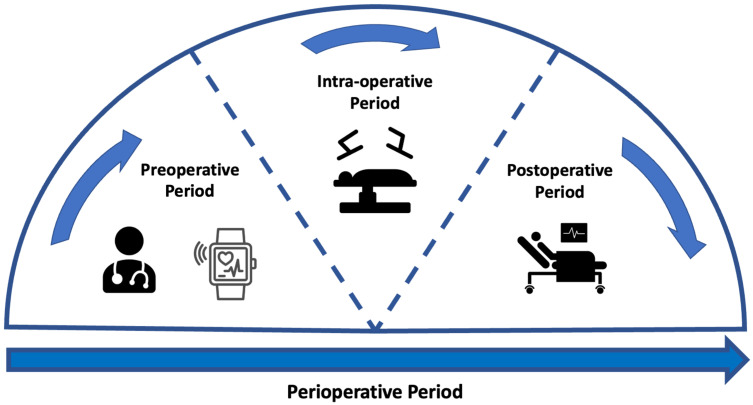
Figure to show the stages across the perioperative period. The perioperative pathway refers to the period that spans from the first point at which surgery is considered as a treatment option up until the full recovery [19]. This pathway has several sub-stages [20]. The preoperative period represents the period prior to surgery where any preoperative assessment takes place. The intra-operative period is representative of the period whilst the patient is undergoing treatment. The postoperative period relates to any period immediately following the operation and can continue after patient discharge.

**Figure 2 sensors-24-00482-f002:**
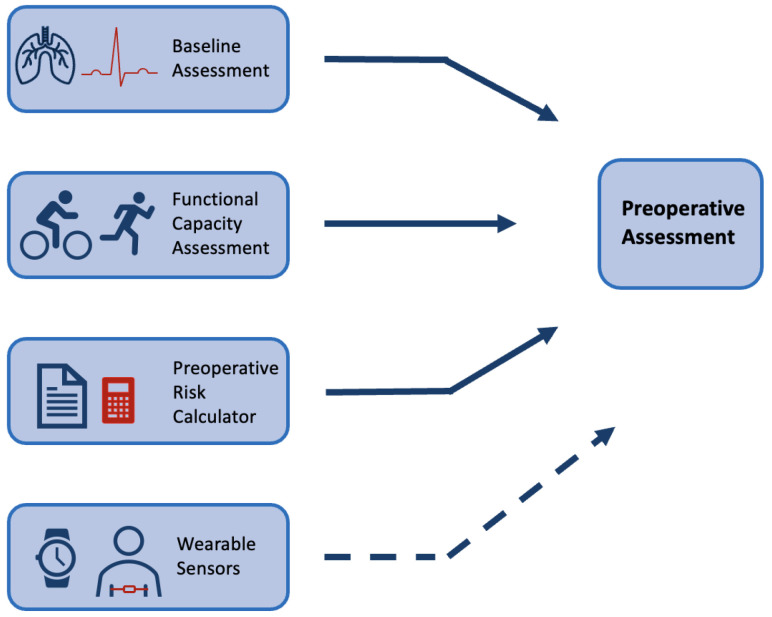
Figure to show common preoperative assessment tools used in practice. The top three boxes present common forms of preoperative assessment that are regularly used in practice (see Section 1.1), whilst the last box with a dashed arrow is included to show the potential for wearable sensors to be used alongside common methods in this context.

**Figure 3 sensors-24-00482-f003:**
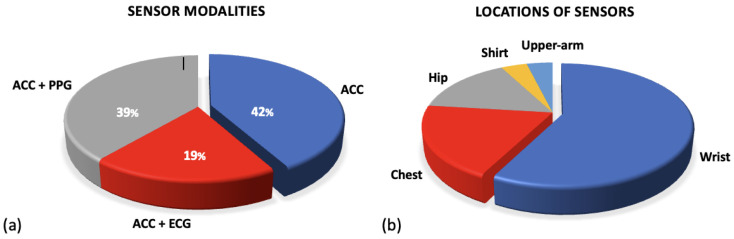
Figures for sensor modalities. (**a**) Shows the percentages of sensor modalities used across research. All research employed accelerometer sensors but a further subsection combine this with either ECG or PPG sensors. (**b**) Figure to show the variations in locations of sensor types. The common locations for sensors used in research applicable to the preoperative period are outlined.

**Figure 5 sensors-24-00482-f005:**
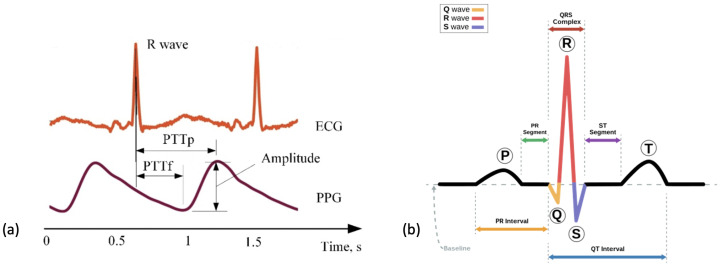
Figures to show the recordings produced from ECG and PPG recordings. (**a**) A comparison of the cardiac signals produced from a PPG versus ECG sensor over a period of 2 s. This figure was produced by Elgendi et al. (2019) and was taken from a larger figure with no changes made as part of the Creative Commons Attribution International 4.0 License [92,100]. (**b**) A segment of an ECG graph that has been portioned to show the stages in a normal cardiac cycle including the P wave, the QRS complex and the T-wave.

**Figure 6 sensors-24-00482-f006:**
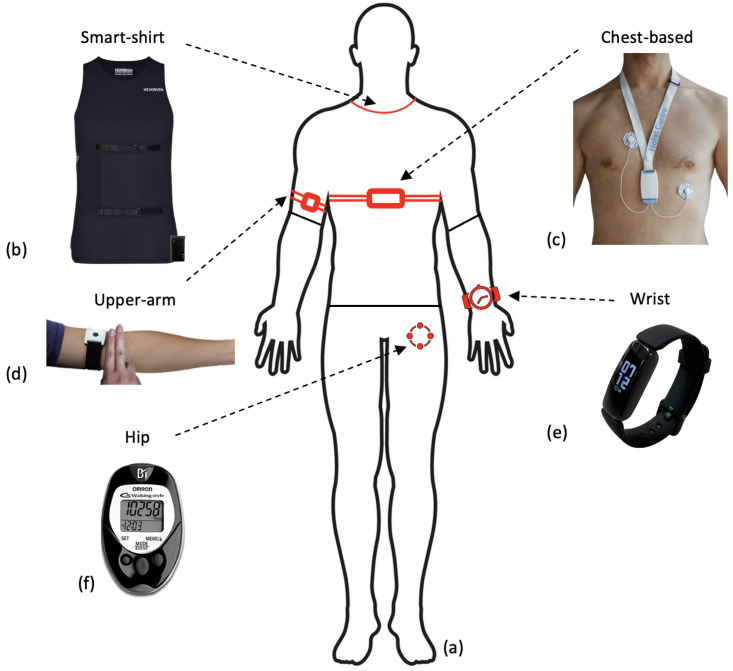
Figure to show wearable sensor devices used in research across the body (**a**) and where these are located (**b**–**f**). (**b**) The Hexoskin smart shirt that collects both ECG and activity data, used with permission from Hexoskin [117]. (**c**) An ECG wearable device that collects recordings from a single-lead ECG device and 3D-accelerometer data, used with permission from [118]. (**d**) An upper-arm PPG sensor utilising reflective PPG detection, similar to that used in preoperative monitoring research [119]. The figure is taken as part of a larger figure from Wang et al. (2023), Creative Commons Attribution International 4.0 License [92,120]. (**e**) The Fitbit Inspire collects a combination of accelerometer and PPG data from the user and is commonly used in preoperative research. The figure is taken from Li et al. (2023), Creative Commons Attribution International 4.0 License [92,121]. (**f**) The OMROM walking style pedometer that utilises a tri-axis accelerometer to collect step data, used in predicting VO2 max [78]. This figure is taken from Bartlett et al. (2017) as part of a larger figure, Creative Commons Attribution International 4.0 License [92,122].

**Figure 7 sensors-24-00482-f007:**
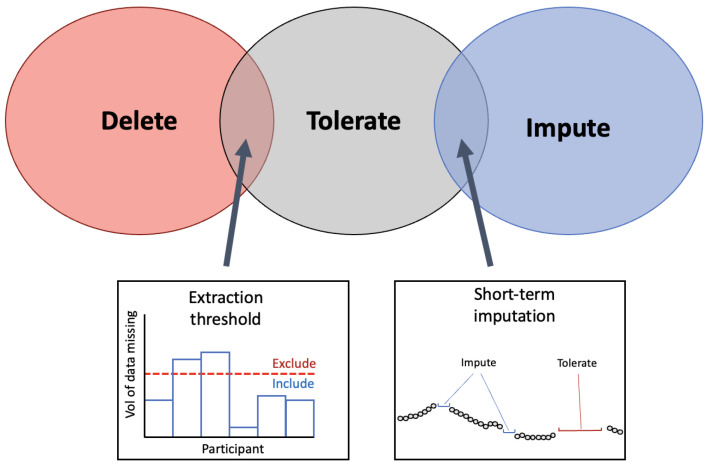
Venn-diagram to present the common methods for handling missing data from WS. The three techniques identified in the literature for handling the missing periods of data are presented in the Venn diagram. At the intersection between ‘delete’ and ‘tolerate’ the implementation of an extraction threshold was identified to delete data below the threshold and tolerate missing data above the threshold. At the intersection between ‘tolerate’ and ‘impute’, imputation on short-term segments of missing periods was identified as a solution that employs that imputation on select segments.

**Figure 8 sensors-24-00482-f008:**
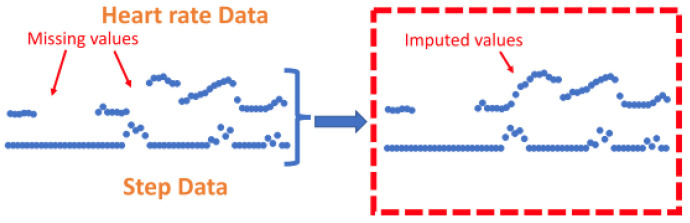
Figure to show imputation using K-nearest-neighbours. Zhang et al. (2023) utilise the KNN technique to impute on short-term segments of missing data under 10 min in length by utilising previous values from both the step count and heart rate signals to calculate missing values. This figure was produced by Zhang et al. (2023) and was taken from a larger figure but had no changes made, taken as part of the Creative Commons Attribution International 4.0 License [71,92].

**Figure 9 sensors-24-00482-f009:**
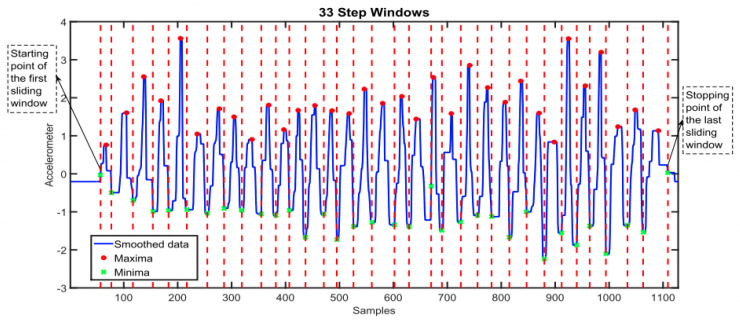
Graph to show the implementation of a maxima and minima step-counting algorithm that counts step number based on the number of steps windows detected. Each red line indicates the stopping point of each step window and the start of the next corresponding window, the length of time between each red vertical line indicates the step window size [142]. This figure was produced by Ho N et al. (2016) and was taken with no changes made, used as part of the Creative Commons Attribution International 4.0 License [92,142].

**Figure 10 sensors-24-00482-f010:**
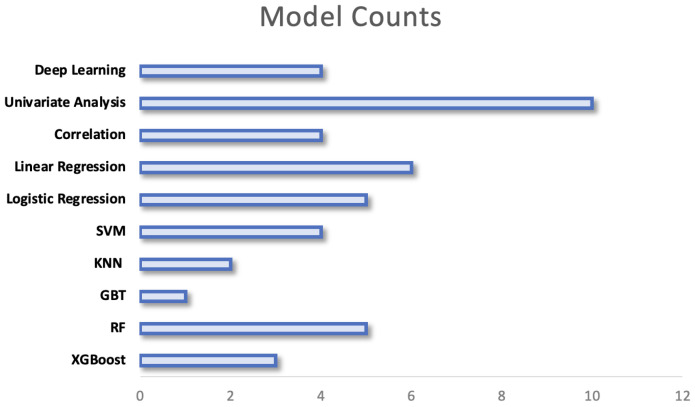
Figure to show the prevalence of each model of analysis across research. In research where multiple models are compared, all models are counted.

**Table 1 sensors-24-00482-t001:** Combination of search terms used for the review of the literature. These searches were combined with Boolean operators and entered into the databases MEDLINE and Web of Science in equal format. Initial investigation of search terms was completed to find the combination of terms that returned optimal results. A narrative review of results is completed in this paper.

Surgery	Preoperative Assessment	Wearable Sensor
major surgery	preoperative	wearable technology
general surgery	pre-surg *	wearable activity monitor
abdominal surgery	preoperative evaluation	heart rate monitor
elective surgery		accelerometer
		fitness tracker
		wearable fitness *

* The asterisk is used for truncation in the search, the asterisk added to end of a term allows the databases to search for all forms of the word to broaden the search.

## Data Availability

No new data were created or analyzed in this study. Data sharing is not applicable to this article.

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
