# Peer review of "Wearable Sensors as a Preoperative Assessment Tool: A Review"

_sensors, 2024, doi:10.3390/s24020482_

Round 1
Reviewer 1 Report
Comments and Suggestions for Authors
This review article explores the potential of wearable sensors as a preoperative assessment tool, analyzing their application in monitoring patients before surgery. It highlights the prevalent use of accelerometers in conjunction with other sensors, discussing data processing methods and the limitations posed by proprietary algorithms in commercial devices. The presented review offers recommendations for future research to enhance WS as a reliable preoperative assessment tool. This review article holds considerable value and boasts a well-organized structure, making it suitable for acceptance in its current form for publication in the journal of Sensors. Some issues illustrated below could be helpful for enhancing the quality of this review.
1. The advantages and limitations associated with using accelerometers as the sole sensor modality in wearable sensors studies versus combining them with photoplethysmography or electrocardiography sensors should be highlighted in the article.
2. How do commercially available sensors, which rely on internal proprietary algorithms, affect the extraction of features and the preprocessing of data in wearable sensor research, particularly when assessing cardiac activity? What are the limitations imposed by such sensors in extracting only heart rate as a feature from electrocardiography sensors within the literature?
3. The challenges and complexities associated with integrating deep learning methodologies within preoperative wearable sensor research should be explored in the outlook and challenges.
Reviewer 2 Report
Comments and Suggestions for Authors
The authors studied the reliability of wearable sensors as preoperative evaluation from the perspective of wearable sensors. In this paper, from the perspectives of hardware sensing technology, signal processing technology, feature output and data analysis, the author analyzes the specific division of wearable sensors in the field of preoperative evaluation in detail, and puts forward reasonable suggestions for the challenges and opportunities faced by wearable sensors in the future. Overall, the content of the article is well-organized, but there are still several problems that need to be improved:
1. In 2.1, can the accelerometry be added to explain the relationship in detail between the increase in step count and all-cause mortality, cardiovascular disease (CVD)?
2. Can you explain in detail the corresponding relationship between PQRST in ECG and human health in Figure 5?
3. In terms of missing data, could you discuss the solution to the problem that leads to data missing, such as poor electrode placement or contact with skin alongside removal of device?
4. Add more detailed literature research to further illustrate the solution for noise and encoding time.
5. In terms of feature extraction, various test functions should be added for the specific impact of preoperative assessment.
6. Some charts and graphs should be added to the data analysis section to further explain.
7. Whether battery problems and wearable sensors have some relationship to the preoperative assessment, if so, the relationship between battery problems and wearable sensors should be explained.
8. I suggest the authors to give a more comprehensive reference review in recent significant improvement in the field of flexible sensors (such as literatures in Sci. Adv. 8, eade0720 ,2022; Electronics 2022, 11, 1651ï¼›
Comments on the Quality of English LanguageMinor editing of English language required
Reviewer 3 Report
Comments and Suggestions for Authors
I appreciate the effort made by the authors and provide valuable insights in summary and review of the most recent works in using wearable sensing for preoperative assessment tools. Overall I think this paper is well presented and here are some of my concerns. First, this review paper could provide a detailed comparison of datasets used in related works, focusing particularly on the number of subjects and patient demographics. This comparison is vital for understanding the scope and applicability of wearable sensor technologies in various clinical settings. A comprehensive dataset comparison would enable readers to better assess the representativeness and generalizability of the findings. Detailing patient demographics, such as age, gender, ethnicity, and health conditions, could provide more insights into data variability and potential biases.
For a review paper, including a comprehensive comparison table summarizing key aspects of the studies reviewed would be beneficial. This table should categorize and summarize different sensor types, signal processing techniques, features extracted, and models developed across the paper(for each section/category). A summary table would enhance clarity and readability, helping readers quickly understand the similarities and differences among the studies.
Regarding studies that utilize wearable sensors as a preoperative assessment tool, important factors include Clinical Validation and Regulatory Considerations, as well as Ethical and Privacy Considerations. If any related works discuss or study these aspects, including this information would significantly benefit the paper. Addressing the clinical validation process, including necessary clinical trials and studies for establishing efficacy and safety, and outlining the regulatory pathways for approval and integration into healthcare settings, is essential. Additionally, the paper should discuss the ethical implications and privacy concerns of using personal health data, including data security, patient consent, and compliance with privacy regulations. These aspects are fundamental to ensuring patient trust and ethical deployment of these technologies in healthcare, particularly in real-world applications and clinical settings.
Similarly, a section on the review of the related works’ cost and benefit analysis of wearable sensor technologies is needed if they exist. This section could detail the costs associated with the development, implementation, and maintenance of these technologies, comparing them with potential benefits like improved patient outcomes, reduced hospitalization times, and overall healthcare cost savings.
